# LH/hCG Regulation of Circular RNA in Mural Granulosa Cells during the Periovulatory Period in Mice

**DOI:** 10.3390/ijms241713078

**Published:** 2023-08-23

**Authors:** V. Praveen Chakravarthi, Wei-Ting Hung, Nanda Kumar Yellapu, Sumedha Gunewardena, Lane K. Christenson

**Affiliations:** 1Department of Cell Biology and Physiology, University of Kansas Medical Center, 3075 HLSIC, 3901 Rainbow Blvd., Kansas City, KS 66160, USA; praghavulu@kumc.edu (V.P.C.); weitinghung@ntu.edu.tw (W.-T.H.); sgunewardena@kumc.edu (S.G.); 2Department of Biostatistics and Data Science, University of Kansas Medical Center, Kansas City, MO 66160, USA; nyellapu@kumc.edu

**Keywords:** circular RNA, granulosa cells, ovulation, post-transcriptional gene regulation, miRNA

## Abstract

Ovarian follicles undergo a series of dynamic changes following the ovulatory surge of luteinizing hormone including cumulus expansion, oocyte maturation, ovulation, and luteinization. Post-transcriptional gene regulatory events are critical for mediating LH follicular responses, and among all RNA isoforms, circular RNA (circRNA) is one of the most abundant forms present in cells, yet they remain the least studied. Functionally, circRNA can act as miRNA sponges, protein sponges/decoys, and regulators of transcription and translation. In the context of ovarian follicular development, the identity and roles of circRNA are relatively unknown. In the present study, high throughput RNA sequencing of granulosa cells immediately prior to and 4-h after the LH/hCG surge identified 42,381 circRNA originating from 7712 genes. A total of 54 circRNA were identified as differentially expressed between 0-h and 4-h time points (Fold Change ± 1.5, FDR ≤ 0.1), among them 42 circRNA were upregulated and 12 circRNA were downregulated. All differentially expressed circRNA between the 0-h and 4-h groups were subjected to circinteractome analysis and identified networks of circRNA-protein and circRNA-miRNA were further subjected to “micro-RNA target filter analysis” in Ingenuity Pathway Analyses, which resulted in the identification of miRNA targeted mRNAs. A comparison of these circRNA target mRNAs with LH-induced mRNAs identified Runx2, Egfr, Areg, Sult1el, Cyp19a1, Cyp11a1, and Hsd17b1 as targets of circKif2, circVcan, circMast4, and circMIIt10. These newly identified LH/hCG-induced circRNA, their target miRNA and protein networks provide new insights into the complex interactions associated with periovulatory follicular development.

## 1. Introduction

Ovarian follicles undergo dramatic developmental and metabolic changes during the periovulatory period. During this time, the ovulatory surge of luteinizing hormone (LH) triggers a number of molecular and cellular events to initiate cumulus expansion, oocyte maturation, ovulation, and luteinization of the follicular wall [1]. LH-mediated signal transduction and transcriptional changes have been well documented, while post-transcriptional events have been studied less. Post-transcriptional regulatory mechanisms can involve different RNA isoforms (e.g., microRNA, snRNA, and lncRNA) in addition to alternative splicing, changes in localization, and translational control. The regulatory role played by non-coding RNA isoforms in periovulatory follicular development remains woefully under explored. Among these, circular RNA (circRNA) are the least investigated even though they are the predominant non-coding isoform, with exception of ribosomal RNA present in humans and mice [2].

Circular RNA are covalently closed RNAs and are generated by lariat-driven circularization, cis-acting elements such as Alu/Sine/Line elements, trans-acting splicing factors, and RNA binding proteins (RBP) that direct back splicing [3]. Recent advances in sequencing technologies and bioinformatic tools have led to findings that circRNA are evolutionary conserved, transcriptionally active (some), and play major roles in cellular metabolism [4]. Circular RNA are stable molecules in comparison to mRNA due to the lack of a 5′ end and 3′ poly A tail [3], and therefore are not susceptible to 3′-5′exonucleases. Functionally, circRNA have been shown to regulate parental (cis) gene expression through multiple processes including epigenetic DNA methylation [5], competition with parental gene expression as back splicing competes with normal splicing [6], and interference with RNA polymerase II activity by forming a complex with U1snRNPs and promoting the transcription of its parental gene [7]. However, the most common mechanism tied to circRNA is its action as a miRNA sponge [8], where a circRNA binds to the miRNA, presumably in the context of the miRNA-induced silencing complex, thereby reducing the availability of the miRNA and causing the upregulation of the miRNA target genes. Similar to the miRNA sponge concept, circRNA can also serve as protein decoys, effectively sequestering RBPs, or forming scaffolds and recruiters to manipulate gene expression [7,9]. Furthermore, in rare cases, some circRNA have internal ribosomal entry sites, and in conjunction with N6-methyladenosine mediated cap independent initiation are capable of being translated [10].

Interestingly, circRNA have been widely implicated in cell proliferation, epithelial-mesenchymal transitions, pluripotency, and early lineage differentiation [11]. Moreover, abnormal circRNA expression has been associated with a variety of diseases including cancer as well as neurological and cardiovascular disorders [12,13,14,15,16]. In the context of female reproduction, very little is known. The presence of circRNA and their interacting miRNA and proteins were analyzed in high and low fertility sows during the follicular and luteal phases of the estrous cycle; and FOXO, TGFβ, and Wnt signaling pathways were identified as downstream targets of circRNA [17]. In women undergoing assisted reproduction technology (ART), circRNA were found to be associated with gamete, embryonic quality, and ART success [18]. Women with primary ovarian insufficiency exhibited altered expression of circRNA in granulosa cells, and circRNA-miRNA network analyses identified the FOXO signaling pathway [19]. Additionally, circRNA profiles in granulosa cells of women suffering from polycystic ovarian syndrome were found to be related to pathways involved in inflammation, proliferation, and vascular endothelial factor signaling [20]. Our current understanding of circRNA within the periovulatory ovarian follicle is poorly understood, yet many of the physiologic events linked to circRNA function occur during the ovulatory period. Thus, we hypothesize that circRNA are likely key regulators within the periovulatory follicle during the ovulatory process. The aim of the present study is to analyze circRNA expression and functional networks in granulosa cells at two time points after gonadotropin stimulation. Detailed analyses of circRNA during the highly important periovulatory period of follicular development might reveal new therapeutic targets to address infertility or block fertility.

## 2. Results

### 2.1. Circular RNA Isoforms in Mural Granulosa Cells

Mural granulosa cells contained 42,381 different circRNA originating from 7712 genes based on the presence of a back spliced isoform (Appendix A). Based on counts per million (CPM) values, circRNA with CPM ≥ 100 in at least three of the six samples reduced the number of circRNA to 1658 isoforms originating from 1077 genes (Appendix A). The distribution of these more abundant circRNA indicated that most circRNA were between 68–1000 bp in length, based on the site of the back spliced exons (Figure 1A), with most circRNA containing between two to five exons (Figure 1B). PCA plot shows four distinct clusters of samples, with circular and linear RNAs segregation shown on the first dimension (PC2), and 0-h and 4-h segregation shown on the second dimension (PC3) (Appendix A).

### 2.2. circRNA vs. Its Linear RNA Form in Granulosa Cells before (0-h) and after (4-h) LH/hCG Surge

Heatmaps of the abundant (1658) circRNA and their linear isoforms were consistent across the triplicate samples at the 0-h and 4-h time points (Appendix A).

Of the abundant (1658) circRNA evaluated at 0-h post-hCG, 791 circRNA were found at higher or in equal counts than their linear RNA isoform (fold change ≥ 1) and 867 circRNA were found to have lower counts than the linear RNA isoforms (Figure 2A; Appendix A). Of the abundant (1658) circRNA evaluated at 4-h post-hCG, 813 circRNA were detected at higher or in equal counts than the linear RNA isoforms (Fold change ≥ 1) and 845 were found at lower counts than the linear RNA isoforms (Fold change < 1) (Figure 2B; Appendix A).

circRNA expressed at equal or higher counts than their linear RNA isoform (Fold change ≥ 1) were subjected to IPA analysis (using native gene names). IPA analysis at the 0 h time point identified different pathways, such as NGF, FGF, TGF-β, ERK/MAPK, JAK/STAT, PTEN, and 12 other signaling pathways (Figure 3A). IPA analysis of 4-h post-hCG granulosa cells identified 11 pathways, including NGF, FGF, TGF-β, cAMP, ERK/MAPK, and the BMP signaling pathways (Figure 4A). Overall, the graphical summary of IPA analysis in both the 0-h and 4-h groups showed that cell cycle progression, cell proliferation, cell survival, ESR1 signaling, NGF signaling, and growth of the embryo were the main mechanisms, where the circRNA (their native genes) play a role (Figure 3B and Figure 4B). Upstream IPA analysis at the 0-h and 4-h post-hCG identified a group of highly expressed circRNA compared to their linear transcripts involved in the ESR1, estrogen (β-estradiol, ESR1), KCNJ2, and miR-144 signaling pathways (Figure 3C and Figure 4C). Among them, *Lhcgr*, *Tgfbr*, *Map2k1*, *Bmpr1b*, and *Fgfr2* were well-established ovarian genes with a prominent role in ovarian follicular development and ovulation (Figure 3C and Figure 4C). As circRNA are highly stable compared to their linear transcripts, once produced, they may remain for extended periods of time in the cells. Therefore, the circRNA observed prior to the LH surge may still be elevated 4-h after the LH surge, leading to their retained appearance.

### 2.3. Differentially Expressed circRNA in Mural Granulosa Cells before and after the LH/hCG Surge

Heatmaps of the differentially expressed circRNA were consistent across the triplicate samples between the 0-h and 4-h time points (Appendix A). Fifty-four circRNA transcripts with a ≥ ±1.5-fold change (FDR ≤ 0.1) were identified as different between the 0-h and 4-h post-hCG in mural GCs (Figure 5A,C). Among them, 42 circRNAs are upregulated (fold change ≥ 1.5, FDR ≤ 0.1) and 12 circRNAs are downregulated (fold change ≤ 1.5, FDR ≤ 0.1). IPA analysis (native gene names) for these 54 differentially expressed circRNA identified Hippo, DNA methylation, VEGF, IL-15, ferroptosis, epithelial adherens, mRNA degradation, and vitamin C transport pathways (Figure 5B). Upstream IPA analysis using the native gene names of the differentially expressed circRNA identified EGFR, VCAN, miR-144, and forskolin signaling pathways (Figure 5D). Of the 54 LH-regulated circRNA, 10 circRNA (i.e., Rnf180, Crim1, Slc7a11, Vcan, Ano4, Slc7a8, Gm20459, Palm2, Ywhae, and Emg1) were found to be of higher expression than their linear isoform, highlighted with arrow marks in Figure 5C.

### 2.4. miRNA and Protein Interaction Network of Selected circRNA

All differentially expressed circRNA between the 0-h and 4-h groups were subjected to circinteractome analysis (https://circinteractome.nia.nih.gov/). Using the matching human circRNA sequence to our identified mouse circRNA sequences resulted in a network of mouse circRNA/corresponding human circRNA/miRNA and a protein network (Appendix A). Those with miRNA or protein interaction sites of three or greater are shown in Table 1. It is hypothesized that circRNA, having a relatively high number of binding sites for a single RBP or miRNA, could act as a sponge or decoy of that RBP or miRNA; we chose a cut-off value as ≥ three sites [21]. In total, 266 interacting proteins were identified; when restricted to those with ≥ three binding sites/circRNA, the number dropped to 132 interacting proteins. Removing duplicates across all of the circRNA resulted in a total of 23 proteins (i.e., FUS, HuR, IGF2BP1, IGF2BP2, IGF2BP3, LIN28A, AGO2, EIF4A3, PTB, FMRP, AGO1, CAPRIN1, DGCR8, EWSR1, ZC3H7B, FXR2, LIN28B, AGO3, C22ORF28, SFRS1, TIAL1, U2AF65) as interacting with our circRNA.

Repeating the ≥ three binding sites/circRNA analyses and removing duplicates, we identified 82 miRNA that interacted with LH-regulated circRNA. These miRNA were subjected to “micro RNA target filter analysis” in IPA, which resulted in a list of 11,040 target mRNAs (Appendix A). These target mRNAs were compared with LH-induced genes (4-h linear RNA versus 0-h linear RNA, Appendix A), and 689 LH-induced genes, targets for these miRNA, were found (Appendix A). Further to the previous literature, we selected some of the LH-induced and ovulation-related genes (i.e., Runx2, Egfr, Areg, Ereg, Pgr, Pge2, Ptx3, Hif, Tnfaip6, Sult1e1, Edn2, Snap25, Adamts1, Smad1, Has2, Vscn, Strp4, Il6, Edn2, Pparα, Cyp11a1, Cyp19a1, Star, Lhcgr, and Fshr [22,23,24,25]) and screened these genes from the miRNA-targeted LH-induced genes; we determined Runx2, Egfr, Areg, Sult1el, Cyp19a1, Cyp11a1, and Hsd17b1 (Figure 6).

## 3. Discussion

Mural granulosa cells contained large numbers of circRNA isoforms originating from a large number of (+7000) genes, and when further curated based on at least 100 counts per million, this number of circRNA still remained at a robust level. When mural granulosa cell circRNA were compared to their corresponding linear mRNA form, over 50% were found to exhibit levels exceeding that of the linear form. One might interpret this as evidence that circRNA are critical players in post-transcriptional gene regulation. However, it is also important to recognize that circularized RNA molecules are more resistant to exonucleases, and thus their presence alone could simply be a result of stochastic RNA synthesis and degradation pathways. In addition, we determined that the LH surge elicited a rapid change (within 4-h) in the abundance of a small number of RNA transcripts, most of which increased. Subsequent bioinformatic analyses of these selected circRNA indicated that they had the ability to bind to a small number of enriched proteins and miRNA partners related to processes well known to be critical in the growth/development and maturation of the ovarian follicle [22,23,24,25,26,27,28,29,30,31,32,33,34,35,36,37,38,39,40]. These initial studies open up a new area of scientific investigation regarding understanding the molecular mechanisms underlying the ovulatory gonadotropin-induced ovarian surge, which might ultimately help us understand human fertility and infertility.

circRNA molecules have many modalities through which they might modify cellular function [5,6,7,8,9,10]. While the most prominently discussed in the literature is clearly the ability to sponge miRNA [8], this ability relies on the presence of multiple binding sites for a specific miRNA in order that a single circRNA could bind sufficient miRNA to elicit a biological effect. In our study, 10 out of 54 differentially expressed circRNA had ≥ three potential miRNA binding sites for any one specific miRNA. Similarly, the ability to sponge proteins is a common mechanism associated with circRNA activity; our studies detected that a relatively small number of proteins, mostly RBPs, were associated with the LH-regulated circRNA. Changes in the expression of these RBPs’ access to linear RNA forms could easily elicit profound effects on cellular behavior. Interestingly, the linear isoforms of the LH-regulated circRNA impacted cellular pathways related to EGFR, VCAN, miR-144, and forskolin (cyclic AMP signaling pathway). The EGFR signaling pathway is known to be critical in cumulus expansion and a facilitator of ovulation [26]. Versican (VCAN) is expressed in mural granulosa cells and is a component of the extracellular matrix and serves in ovulation by remodeling the matrixes [41]. miR-144 regulates mural granulosa cell apoptosis through the CP2/miR-144/COX-2/PGE2/ovulation pathway in mGCs [27]. Lastly, the cAMP (forskolin) pathway is the well-established primary cell signaling pathway through which LH/hCG mediates its actions upon the binding of the LH receptor [28,42]. LH-induced circRNA with circRNA counts were higher than that of linear counts, found through a comparative analysis with corresponding linear RNA; i.e., Rnf180, Crim1, Slc7a11, Vcan, Ano4, Slc7a8, Gm20459, Palm2, Ywhae, and Emg1.

The functional relevance of all LH-induced circRNA was determined through circinteractorme studies. This resulted in a list of RBPs and interacting miRNAs. The RBPs identified have known roles in ovarian physiology; for instance, AGO2 is essential for early embryonic development; i.e., the destruction of oocyte transcripts and the activation of zygotic transcripts [29]. IGFBP1 expressed in granulosa cells is known to be involved in follicular growth and ovulation [30]. ELF4A3 is a component of the exon junction complex and is associated with mRNA exportation, cytoplasmic localization, and non-sense-mediated decay [43]. HUR binds to AU-rich regions of mRNA and stabilizes the mRNA [31]; HUR is reported to be essential for embryonic and extra-embryonic development [32]. Polypyrimidine tract-binding protein PTB or hnRNP I is expressed in the oocytes, granulosa cells, and stroma cells [33], and is involved in splicing, RNA stability, and localization [34]. FMNRP is expressed in granulosa cells and oocytes [35] and permutations in the *Fmr1* (gene coding FMNRP) lead to sub-fertility in female mice [35]. FMNRP binds mRNA and regulates protein synthesis through ribosomes [36]. DGCR8 is involved in miRNA biogenesis; a loss of DGCR8 in *C. elegans* leads to a failure to ovulate and infertility [37]. An overexpression of LIN28A in cultured human granulosa cells decreases estrogen levels, ATP content, and causes mitochondrial dysfunction [38]. IGF2BP1 regulates GCs viability, cell cycle, and cell proliferation through the m6A-dependent mRNA stability of the MDM2 gene [39]. IGF2BP2 regulates the transcription and alternate splicing of genes involved in follicular development, such as MBD3, FN1, TFDP1, and MKNK2 [40]. In total, the identified RBPs associated with the identified mural granulosa cell circRNA have a robust and well-established role in ovarian physiology.

The functional relevance of all miRNA interacting with LH-induced circRNA was determined by microRNA filter analysis. A network of miRNA and their target mRNA was identified. This list was compared with LH-induced genes identified from our studies and previous reports on granulosa cell differentiation, cumulus expansion, and ovulation [22,23,24,25]. Thus, we found CircMast4(Gm6211)/hsa-miR-323-3p/EGFR, CircMast4(Gm6211)/hsa-miR-155/HSD17B12, CircMast4(Gm6211)/hsa-miR-155/SMAD1, CircMast4(Gm6211)/hsa-miR-361-3p/CYP19A1, CircMast4(Gm6211)/hsa-miR-377/AREG, CircMast4(Gm6211)/hsa-miR-877/SULT1E1, CircVcan/hsa-miR-326/FSHR, CircMllt10/hsa-miR-647/RUNX2, CircKif24/hsa-miR-296-5p/CYP11A1, and CircMllt10/hsa-miR-647/RUNX2. Runx2, Sult1e1, Cyp19a1, Cyp11a1, and HSD17B12 play roles in steroidogenesis [22,23,25], Runx2 and Areg regulate cumulus expansion [22,24], Smad1 regulates luteinization [22], FSHR is a key receptor in FSH signaling that leads to ovarian follicular development [44]. Thus, LH-regulated circRNA regulates the expression of key genes involved in steroidogenesis, cumulus expansion, ovarian follicular development, and luteinization through miRNA sponging.

In conclusion, this work identified a small number of circRNA transcripts that were altered by the LH surge; with the linear forms of some of these genes having clear and established roles in ovarian function. Whether the biogenesis of the circRNA observed here is important for the subsequent actions by the circRNA, through a sponging mechanism or other mechanisms, or whether the biogenesis of the circRNA was merely to reduce expression of linear isoform, remains to be evaluated. The circRNA identified in this experiment provide a foundation for these future functional studies, which will ultimately address the importance of this unique form of RNA in the ovulatory process.

## 4. Materials and Methods

### 4.1. Animals

All procedures involving animals were reviewed and approved by the Institutional Animal Care and Use Committee at the University of Kansas Medical Center and were performed in accordance with the Guiding Principles for the Care and Use of Laboratory Animals. Experiments were performed on wild-type female mice. Mice were maintained in a humidity and temperature-controlled environment with a 14-h light, 10-h dark cycle (7 am to 9 pm) with ad libitum access to food and water; 21-day-old mice were given an intraperitoneal (i.p.) injection of 5 IU of equine chorionic gonadotropin (eCG; Calbiochem, San Diego, CA, USAfollowed by 5 IU of human chorionic gonadotropin (hCG) 46–48 h later. Mice were euthanized at 46-h after eCG alone (0-h), or following eCG + 4-h post-hCG administration. Ovaries and oviducts were collected by dissection distal to the uterotubal junction to ensure the complete removal of the oviduct without any disruption of the ovary.

### 4.2. Isolation of RNA from Mural Granulosa Cells

Ovaries from three wild-type mice euthanized at 0-h and three wild-type mice euthanized at 4-h were used for the isolation of mural granulosa cells. Studies on total transcriptome changes (either upregulation or downregulation) at different time points (0-h, 4-h, 12-h) after hCG showed that approximately 78% of total transcriptomic changes of either upregulation or downregulation occur at 4-h [45]. That is the reason why we focused on the 4-h post-hCG time point in our study, using the 0-h as the control. Ovaries were dissected free of the ovarian bursa and then rinsed in sterile dPBS to remove blood and tissue debris. Ovaries were then placed in dishes of Embryo Max FHM Hepes buffered media with 4 mg/mL BSA. Preovulatory follicles, having defined cumulus granulosa cell layer [46], were visualized under a Nikon SMZ1000 with a Nikon NI-150 high-intensity illuminator (Sanford, NC, USA) and were individually punctured with insulin syringes (28-gauge) to allow for the expulsion of mural granulosa cells (mGC) and cumulus oocyte complexes. Smaller follicles were left with the remaining ovarian tissue, which was removed from the dish before naked oocytes and cumulus oocyte complexes were removed and discarded. Mural GCs and the surrounding media were then collected, placed into a 1.5 mL Eppendorf tube, and gently pelleted at 800× *g*; excess media was aspirated using a pipette and 500 µL of TRI Reagent (Invitrogen, Carlsbad, CA, USA) was applied to the cell pellet. Solubilized cell pellets were frozen at −80 °C overnight followed by the completion of the manufacturer’s RNA extraction protocol (Invitrogen, Carlsbad, CA, USA).

### 4.3. RNA Sequencing and circRNA Expression Analyses

The extracted total RNA was subjected to RNA quality analysis using an Agilent 2100 Bioanalyzer (Santa Clara, CA, USA) and samples with RIN values over 9 were selected for RNA sequencing analyses. RNA sequencing was performed on mGCs collected after only eCG injection (0-h) and 4-h post-hCG (n = 3 for each sample). Total stranded RNA sequencing was performed at a 100-cycle paired-end resolution. Samples were analyzed in biological triplicates. Sequencing generated between 58.3 and 78.7 million reads per sample. The read quality was assessed using FastQC software (v0.11.7) [47] and cleaned using cutadapt software (v1.14) [48]. On average, the per sequence quality score measured in the Phred quality scale was above 30 for all the samples. Circular RNA were obtained using the CIRCexplorer analysis toolset (v1.1.7) [49] with recommended settings. The reads were mapped to the mouse genome (Mus musculus GRCm38) using the splice junction mapper, TopHat, and TopHat-Fusion (v2.1.0) [50] using default parameters. Expression normalization and differential expression calculations were performed using the edgeR software (v3.12.1) [51] to identify statistically significant differentially expressed circRNA. The workflow for circRNA expression analysis is given in Appendix A. A more detailed description of the CIRCexplorer part can be found here: https://circexplorer2.readthedocs.io/en/latest/tutorial/pipeline (accessed on 3 October 2018)

### 4.4. Pathway Analyses

The resultant differentially expressed circular RNA native genes from these groups were subjected to Ingenuity Pathway Analyses to determine the functional pathways in which the native genes are involved. Differentially expressed circRNA compared with their linear mRNA form in the 0-h and 4-h groups (Appendix A) were subjected to IPA analysis, mainly disease and functions and upstream analysis. Similarly, differentially expressed circRNAs between the 0-h and 4-h groups (Appendix A) were subjected to IPA analysis, mainly disease and function and upstream analysis.

### 4.5. miRNA and Protein Interaction Prediction

All LH-induced circular RNA (circRNA 4-h vs. 0-h) were subjected to circinteractme analysis (https://circinteractome.nia.nih.gov/ accessed on 25 May 2023) to predict the interacting proteins and miRNA. Circinteractome is a human database, so initially, we compared the circRNA sequence from our data with human circRNA via BLAST (https://blast.ncbi.nlm.nih.gov accessed on 26 May 2023). Based on the matching and the region of the matching sequence, the miRNA and proteins were identified. Out of 54 circRNA, we found 47 circRNA had matching human circRNA (Appendix A). The miRNA identified from this study were subjected to “micro-RNA target filter analysis” in IPA (IPA; Qiagen Bioinformatics accessed on 05.28.2023). The resultant target mRNAs were compared to LH-induced mRNAs from Appendix A, which provided a list of miRNA that target LH-induced genes. Furthermore, we scanned this list for some marker genes involved in granulosa cell differentiation, cumulus expansion, and ovulation.

### 4.6. Statistics

Three biological replicates were used for each group (0-h and 4-h) for RNA sequencing. Regarding RNA sequencing data, edgeR software [51] was used to identify statistically significant differentially expressed circRNAs. Significance *p*-values were adjusted for multiple hypotheses testing using the Benjamini and Hochberg method [52], establishing a false discovery rate (FDR) for each circRNA.

## Figures and Tables

**Figure 1 ijms-24-13078-f001:**
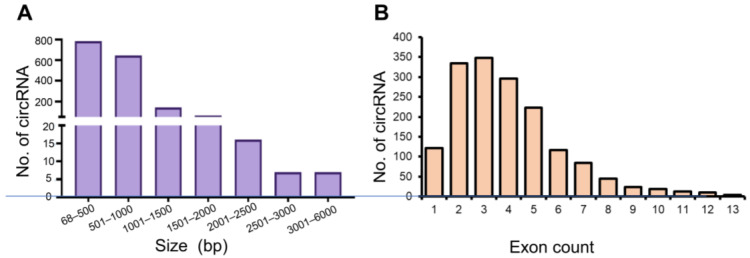
Distribution of the abundant circRNA based on size (**A**) and exon count (**B**) within periovulatory mural granulosa cells from mice treated with eCG alone (for 46 h) and eCG + hCG for 4-h.

**Figure 2 ijms-24-13078-f002:**
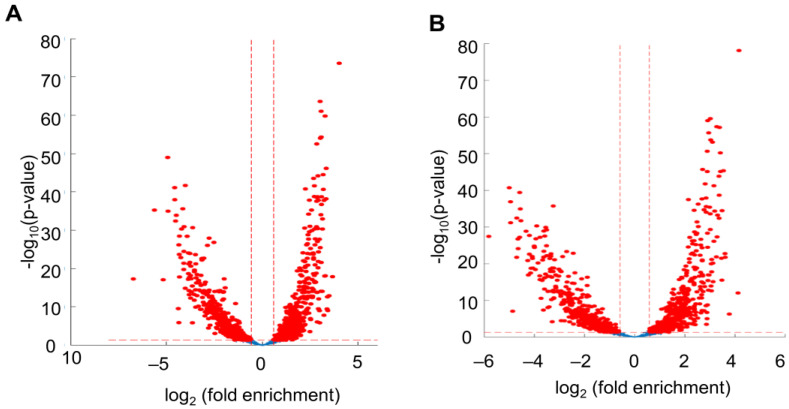
Volcano plots show circRNA compared to their corresponding linear RNA isoforms in mural granulosa cells before and after the LH/hCG surge. (**A**) Volcano plot showing circRNA versus linear RNA with log2FC in X-axis and log 10 *p* Values in Y-axis in 0-h mural GCs. (**B**) Volcano plot showing circRNA versus linear RNA with log2FC in X-axis and log 10 *p* values in Y-axis in 4-h mural GCs.

**Figure 3 ijms-24-13078-f003:**
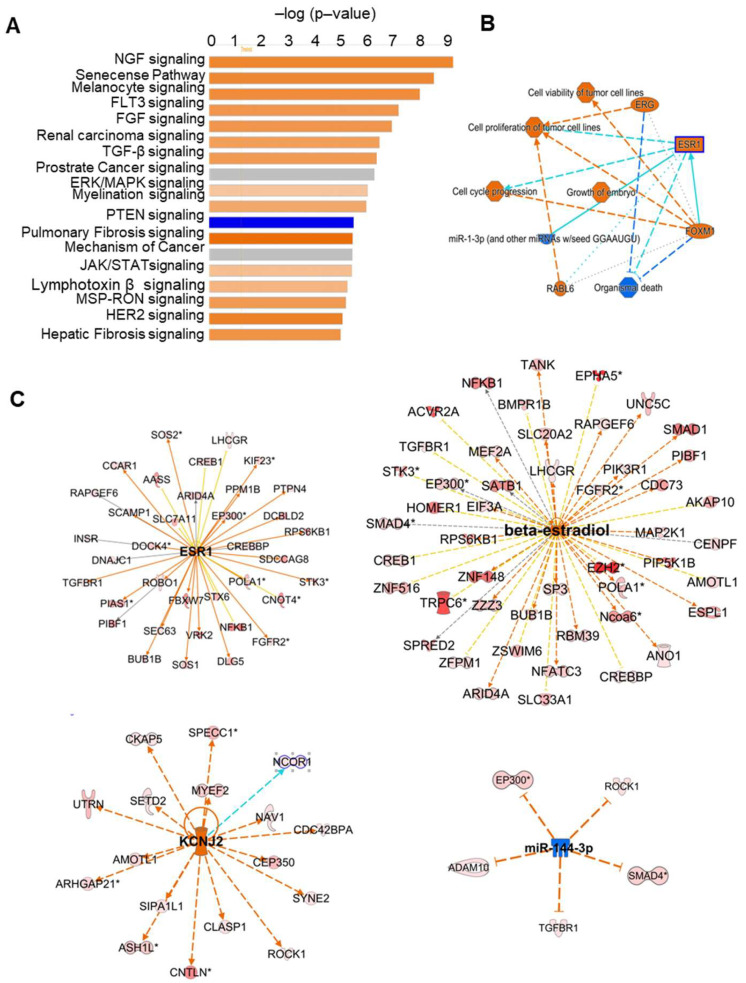
IPA analysis of highly expressed circRNA transcripts compared to their corresponding linear isoforms in 0-h mural granulosa cells. (**A**) IPA analysis of abundant circRNA in 0 h mural GCs showed different pathways such as the NGF, FLT3, FGF, TGF-β, ERK/MAPK, PTEN, and JAK/STAT signaling pathways. (**B**) A graphical summary of IPA analysis showed cell cycle progression, cell proliferation, growth of the embryo, miR-1-3p pathway, ESR1 signaling, ERG signaling, etc., as the main mechanisms where the identified circRNAs (their native genes) are involved. (**C**) Upstream IPA analysis of the highly abundant circRNA compared to their linear RNA isoforms in 0-h group GCs showed higher levels of multiple circRNA, the genes of which are linked to the downstream signaling of ESR1, estrogen (beta-estradiol and ESR1), KCNJ2, and miR144 pathways. * represents duplicates, multiple identifiers in the data set matches to a single gene (multiple circular RNA from a single gene).

**Figure 4 ijms-24-13078-f004:**
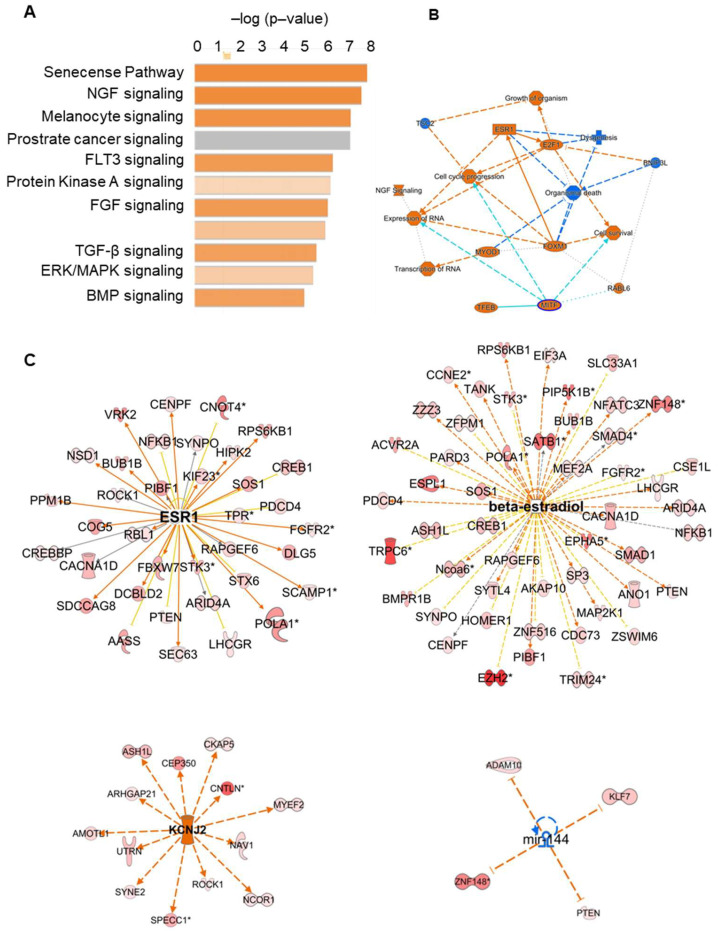
IPA analysis of highly expressed circRNA transcripts compared to their corresponding linear isoforms in 4-h mural granulosa cells. (**A**) IPA analysis of abundant circRNA in 4-h mural GCs showed different pathways, such as the NGF, FLT3, FGF, TGF-β, ERK/MAPK, Protein Kinase A, and BMP signaling pathways. (**B**) Graphical summary of IPA analysis showed cell cycle progression, cell proliferation, miR-1-3p path way, expression and transcription of RNA, ESR1 signaling, etc., as the main mechanisms where the identified circRNAs (their native genes) are involved. (**C**) Upstream IPA analysis of the highly abundant circRNA compared to their linear RNA isoforms in the 4-h group GCs showed higher levels of multiple circRNA, the genes of which are linked to the downstream signaling of the ESR1, estrogen (beta-estradiol and ESR1), KCNJ2, and miR-144 pathways. * represents duplicates, multiple identifiers in the data set matches to a single gene (multiple circular RNA from a single gene).

**Figure 5 ijms-24-13078-f005:**
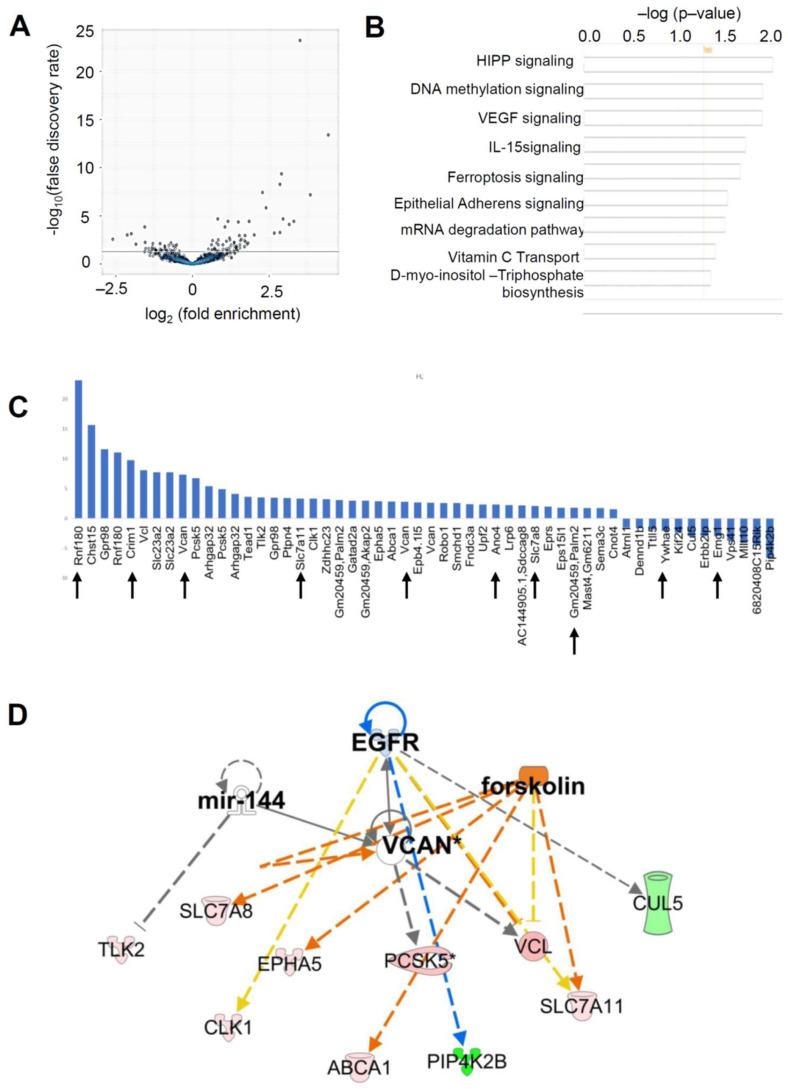
LH (hCG)-regulated circRNA transcripts in mural granulosa cells. (**A**) Volcano plot showing differentially expressed circRNA transcripts only between eCG (0-h) and eCG + 4-h post-hCG in mural GCs. (**B**) IPA analysis of differentially expressed circRNAs with their native gene names between the 0-h and 4-h groups identified different signaling pathways, such as HIPPO signaling, DNA methylation signaling, VEGF signaling, IL-15 signaling, Ferroptosis signaling, Epithelial adherens signaling, mRNA degradation signaling, Vitamin C transport, and D-myo-inositol-Triphosphate biosynthessis signaling pathways. (**C**) Up- and downregulated (FDR ≤ 0.1) circRNA transcripts in mural GCs before and after the LH/hCG surge. circRNA 4-h /linearRNA 4-h vs. circRNA 0-h/linearRNA 0-h was compared with LH-induced circRNA (circRNA 4-h vs. circRNA 0-h) to identify the LH-induced circRNA with circRNA count more than that of the linear count, which are shown with arrow marks. (**D**) Upstream IPA analysis of differentially expressed circRNA between 0-h and 4-h mural GCs showed the upregulation of different circRNA involved in the downstream signaling of the EGFR, miR-144, VCAN, and forskolin pathways. * represents duplicates, multiple identifiers in the data set matches to a single gene (multiple circular RNA from a single gene).

**Figure 6 ijms-24-13078-f006:**
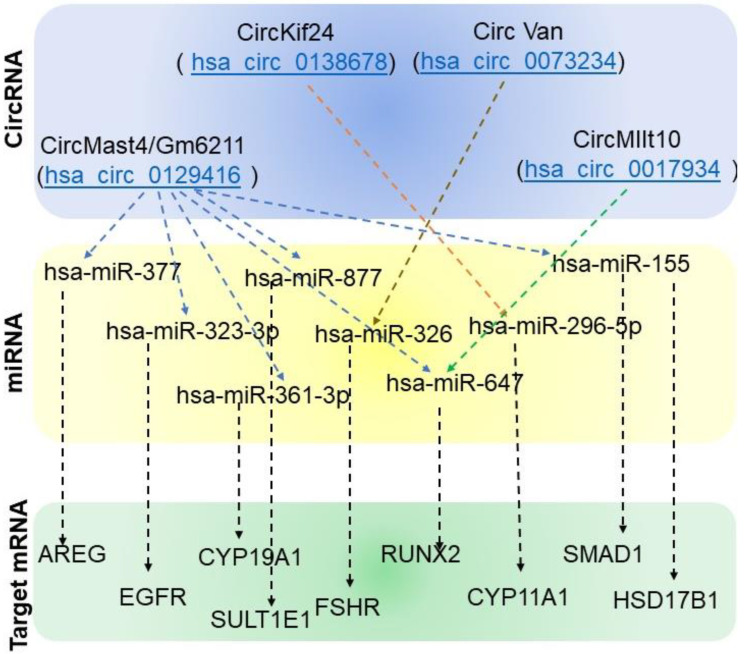
Mouse circRNA (corresponding human circRNA)-miRNA-target mRNA of crucial genes involved in LH-induced periovulatory follicular development. MicroRNA identified from the circinteractome studies identified eight miRNA and nine target mRNAs.

**Table 1 ijms-24-13078-t001:** Table showing list of mice circRNA identified from our studies their corresponding human circular RNA, target miRNAs and interacting proteins as determined from circinteractome anlaysis (https://circinteractome.nia.nih.gov/). This table was further refined with miRNAs and proteins having circRNA binding sites greater than or equal to 3. Note: numbers within the parenthesis indicate the no. of binding sites.

**Gene**	**circRNA Chromosomal Region**	**Corresponding Human circRNA**	**circRNA Interacting Factors**	
			**miRNA**	**Protein**
**Gpr98**	**13:81270744-81283472**	hsa_circ_0073295	hsa-miR:1252 (7); 197 (7)	FUS (6); HuR (15); IGF2BP1 (3); IGF2BP2 (6); IGF2BP3 (6); LIN28A (12).
Crim1	17:78344350-78355177	hsa_circ_0002938		AGO2 (11); EIF4A3 (8); PTB (3).
Slc23a2	2:132089096-132094240	hsa_circ_0059360		AGO2 (3); EIF4A3 (28); FMRP (3); PTB (8).
Vcan	13:89721567-89731534	hsa_circ_0073234	hsa-miR: 326 (3); 330-5p (3); 431 (3); 494 (3); 587 (7); 622 (4); 640 (5); 767-3p (4).	AGO1 (3); CAPRIN1 (3); DGCR8 (5); EWSR1 (6); FMRP (71); FUS (8); HuR (43); IGF2BP1 (41); IGF2BP2 (38); IGF2BP3 (43); LIN28A (33); METTL3 (1); ZC3H7B (6).
Ano4	10:89016075-89035181	hsa_circ_0027873		EIF4A3 (11).
Chst15	7:132262683-132271001	hsa_circ_0000264		AGO2 (6); EIF4A3 (12); FMRP (3); IGF2BP1 (3); PTB (5).
Vcl	14:20982574-20987118			EIF4A3 (3); FMRP (7); LIN28A (3); PTB (3).
Tlk2	11:105240361-105241661	hsa_circ_0008371		EIF4A3 (3).
Ptpn4	1:119715947-119773271	hsa_circ_0117171	hsa-miR: 197 (3); 532-3p (3).	AGO2 (5); EIF4A3 (7).
Slc7a11	3:50417988-50438915	hsa_circ_0070981		AGO2 (4); EIF4A3 (9); FUS (4); HuR (9); PTB (4).
Clk1	1:58417013-58417383	hsa_circ_0057725		AGO2 (6); EIF4A3 (7).
Zdhhc23	16:43973468-43974166	hsa_circ_0066837		EIF4A3 (16).
Gm20459,Palm2	4:57638010-57648120	hsa_circ_0137684		AGO1 (7); AGO2 (30); DGCR8 (10); EIF4A3 (51); EWSR1 (4); FMRP (95); FXR2 (3); HuR (50); IGF2BP1 (20); IGF2BP2 (19); IGF2BP3 (25); LIN28A (6); LIN28B (5); PTB (4).
Abca1	4:53127597-53133015	hsa_circ_0005443		EIF4A3 (3).
Robo1	16:72742126-72742452	hsa_circ_0004788		AGO2 (4); DGCR8 (3); PTB (3).
Smchd1	17:71436706-71448846	hsa_circ_0046712	hsa-miR:1183 (3); 1243 (3); 1245 (1); 1248 (4); 1305 (4); 331-5p (3); 513a-3p (5); 607 (5).	AGO2 (26); EIF4A3 (80); EWSR1 (8); FMRP (8); HuR (10); IGF2BP3 (4); LIN28A (3).
Lrp6	6:134541652-134542045	hsa_circ_0000378		EIF4A3 (5); FMRP (4).
Eprs	1:185386246-185397267	hsa_circ_0007739		AGO2 (6); FMRP (5); HuR (3).
Eps15l1	8:72367904-72380306	hsa_circ_0049885	hsa-miR:1231 (3).	FMRP (3).
		hsa_circ_0137685		EIF4A312 (12).
Mast4,Gm6211	13:103140163-103171556	hsa_circ_0129416	hsa-miR:1178 (3); 1236 (3); 1243 (3); 1245 (3); 1256 (4); 1270 (5); 1299 (3); 1324 (4); 142-5p (4); 145 (3); 155 (3); 186 (6); 194 (5); 31 (4); 323-3p (3); 330-3p (3); 335 (4); 338-5p (4); 361-3p (3); 377 (3); 432 (5); 433 (3); 488 (4); 495 (3); 498 (3); 513a-3p (3); 516b (4); 532-3p (3); 545 (5); 548b-3p (6); 548c-3p (7); 548p (7); 549 (4); 555 (3); 556-3p (3); 569 (3); 576-5p (5); 578 (5); 580 (3); 587 (3); 605 (4); 607 (4); 620 (5); 623 (5); 633 (3); 646 (3); 649 (4); 668 (3); 877 (3); 888 (3).	
Sema3c	5:17678312-17682083	hsa_circ_0003634	hsa-miR:503 (3).	AGO2 (7); IGF2BP3 (3); EIF4A3 (14); HuR (5); PTB (5).
Cnot4	6:35077981-35080226	hsa_circ_0003629		AGO2 (6); HuR (3).
Atrnl1	19:57751630-57777945	hsa_circ_0092760	hsa-miR:1238 (4); 409-3p (3); 515-5p (3); 518a-5p (3); 526b (3); 527 (3); 548p (5); 567 (3); 577 (4); 579 (6); 607 (7); 671-5p (3).	FUS (6); IGF2BP1 (3); IGF2BP2 (3); IGF2BP3 (7).
Dennd1b	1:139040000-139062971	hsa_circ_0006324	.	AGO2 (4); EIF4A3 (6).
Ttll5	12:85863564-85879475	hsa_circ_0032699		EIF4A3 (15); PTB (3).
Ywhae	11:75751881-75759428	hsa_circ_0041192		AGO1 (24); AGO2 (92); AGO3 (4); C22ORF28 (8); CAPRIN1 (8); DGCR8 (5); EIF4A3 (20); FMRP (20); FUS (8); HuR (16); IGF2BP1 (16); IGF2BP2 (16); IGF2BP3 (11); LIN28A (11); LIN28B (6); METTL3 (2); MOV10 (3); PTB (8); SFRS1 (12); TIA1 (3); TIAL1 (4); TNRC6 (1); U2AF65 (3); ZC3H7B (10).
Kif24	4:41400437-41414999	hsa_circ_0138678	hsa-miR: 1825 (3); hsa-miR-640 (3); hsa-miR-647 (3); hsa-miR-942 (3).	EIF4A3 (12); PTB (4).
Cul5	9:53642518-53642826	hsa_circ_0024169		EIF4A3 (11).
Erbb2ip	13:103866649-103889285	hsa_circ_0001492		EIF4A3 (3).
Emg1	6:124705158-124705656	hsa_circ_0025313		AGO2 (9); EIF4A3 (10); FMRP (7); HuR (8); LIN28A (3).
Vps41	13:18810432-18814293	hsa_circ_0002808		EIF4A3 (6).
Mllt10	2:18122245-18126229	hsa_circ_0017934	hsa-miR:1200 (4); 494 (3); 604 (3); 647 (3).	AGO2 (7); EIF4A3 (25); HuR (8); IGF2BP2 (4); IGF2BP3 (4).
Pip4k2b	11:97726705-97732755	hsa_circ_0043379		EIF4A3 (8); FMRP (4); HuR (5).

## Data Availability

All RNA seq files were submitted to Geo database (GSM7666417-GSM7666422).

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
