# Peer review of "LH/hCG Regulation of Circular RNA in Mural Granulosa Cells during the Periovulatory Period in Mice"

_ijms, 2023, doi:10.3390/ijms241713078_

Round 1
Reviewer 1 Report
The manuscript of V. Praveen Chakravarthi et al. performed an exhaustive characterization of the circRNAs expressed in granulosa cells before and after LH surge, focusing on those differentially expressed between both groups.
Using various bioinformatics tools, the authors identify differentially expressed circRNAs and their potential interactions with other molecules, allowing them to suggest possible roles in ovarian physiology.
The results are presented in a clear, complete, and orderly manner, and provide answers to some of the unresolved questions on this topic. Although the possible functional implications of these findings are still not entirely clear, they constitute a first step towards understanding the role of circRNAs in the complex events that take place during the periovulatory period. Its possible implications are discussed considering a wide spectrum of possibilities so that the discussion is comprehensive, relevant, and up-to-date. The latest advances on the subject have been considered.
Minor comments
Legend of Figure 4, part A- I understand that IPA analysis is referred to circRNA presented in granulosa cells after 4 hs. The same comment is for part C of the figure. Please, revise.
Author Response
Reviewer 1.
Legend of Figure 4, part A- I understand that IPA analysis is referred to circRNA presented in granulosa cells after 4 hs. The same comment is for part C of the figure. Please, revise.
Corrected as suggested.
Reviewer 2 Report
The manuscript addresses LH-dependent regulation of circular RNA and its role in mouse periovulatory follicle development. Although the topic and the results are really interesting and the manuscript is well written, I provide some significant comments below, especially with regard to the description of the methods, which could greatly improve the scientific level of the manuscript.
Introduction – the hypothesis and the aim of the study are missing.
M&M – the authors should clearly state how many biological/technical replicates were studied. The authors mention that they used 21-day-old females, did the analysis of vaginal epithelial cell smears was performed.
What does "large antral follicles" mean - the follicle classification system or some references should be given.
The justification should be given for the choice of a specific incubation period with treatments and treatment doses (eCG and hCG).
Has PCA analysis been performed?
Due to the increasing interest in the circRNA, Authors should clearly specify all the ‘recommended settings’ (line 355) or ‘default parameters’ (line 357) used in circRNA expression analysis. The workflow of the flowchart is necessary for readers who are not familiar with bioinformatic analyses. The justification why the authors presented results of miRNA and proteins with binding sites greater than 3 must be provided.
RNAseq raw data should be provided as the GEO no.
There are some typos and stylistic mistakes.
Author Response
Reviewer 2.
Introduction – the hypothesis and the aim of the study are missing.
Hypothesis and aim of the study were included in the discussion as per the reviewer’s suggestion (Lines 80-83).
M&M – the authors should clearly state how many biological/technical replicates were studied.
The number of mice used in the study and the biological replicate details were given in the revised manuscript (Lines 329-330, 390).
The authors mention that they used 21-day-old females, did the analysis of vaginal epithelial cell smears was performed.
With respect to vaginal smears these mice are immature, thus they are not cycling. It is a well-established procedure to stimulate immature as well as mature mice with PMSG, to promote synchronous development of multiple ovulatory follicles, followed by hCG to mimic the LH surge.
What does "large antral follicles" mean - the follicle classification system or some references should be given.
We have addressed this in the text by noting that we are poking only the large preovulatory follicles and presented appropriate reference in the revised manuscript indicating the classification of follicle. (Line 337).
Justification should be given for the choice of a specific incubation period with treatments and treatment doses (eCG and hCG).
Studies on total transcriptome changes (either upregulation or down regulation) at different time points (0 h, 4 h, 12 h) after hCG showed that approximately 78% of total transcriptomic changes either upregulation or down regulation happens at 4 h time (Shirafuta, Tamura et al. 2021). That is the reason why we have taken 4 h time point in our study and 0 h as control. Respective reference and this justification was given in the revised manuscript. (Lines 330-334)
Has PCA analysis been performed?
PCA plot showing four distinct clusters of circRNA, linear RNA at two time points 0h and 4h was given as Supplementary Fig. 1 in the revised manuscript. (Lines 93-96)
Due to the increasing interest in the circRNA, Authors should clearly specify all the ‘recommended settings’ (line 355) or ‘default parameters’ (line 357) used in circRNA expression analysis. The workflow of the flowchart is necessary for readers who are not familiar with bioinformatic analyses.
The workflow of the circular analysis was given as supplementary Fig. 4 in the revised manuscript (Lines 362-365).
The justification why the authors presented results of miRNA and proteins with binding sites greater than 3 must be provided.
It is hypothesized that circRNA having relatively high number of binding sites for a single RBP or miRNA acts as a sponge or decoy of that RBP or miRNA. So we took a cut of value as ≥ 3 (Dudekula, Panda et al. 2016) (Lines 198-200).
RNAseq raw data should be provided as the GEO no:
We have submitted the RNA seq files to Geo data base and GEO accession numbers were given in the revised manuscript (407-408).
Typos, grammar and stylistic mistakes were corrected.

Round 2
Reviewer 2 Report
Thank you for responding to all my questions.